# Effects of Plant Growth-Promoting Rhizobacteria on the Physioecological Characteristics and Growth of Walnut Seedlings under Drought Stress

**Fangchun Liu [1,†], Hailin Ma [1,*], Binghua Liu [1,†], Zhenyu Du [1], Bingyao Ma [1] and Dawei Jing [2]**

[1] Institute of Resource and Environment, Shandong Academy of Forestry, Jinan 250014, China
[2] College of Ecology, Resources and Environment, Dezhou University, Dezhou 253023, China
* Correspondence: mahlin@163.com; Tel.: +86-531-88557780
† These authors contributed equally to this work.

**Abstract:** Drought is one of the most brutal environmental factors limiting the productivity of fruit trees. The search for new and efficient microorganisms from unexplored environments that can be used to mitigate the negative effects of water stress is an interesting alternative to alleviate the drought stress experienced by plants. This study aimed to determine the effects of PGPR inoculation on the growth and physioecological characteristics of walnut (*Juglans regia*) seedlings under drought stress. A pot experiment was conducted using *J. regia* seedlings with controlled water supplies at different levels (light, moderate, and severe drought stress and control) and with or without inoculation with *Bacillus cereus* L90, a type of PGPR that produces high levels of cytokinins and indoleacetic acid (IAA). Under well-watered conditions, there was no obvious effect of PGPR inoculation on the antioxidant enzyme activities, osmotic adjustment levels, and photosynthetic characteristics of *J. regia*. As the stress intensity increased, *B. cereus* inoculation increased the antioxidant enzyme activities in walnut seedlings and changed their photosynthetic characteristics. However, levels of osmotic adjustment substances were decreased as a result of PGPR inoculation. Regardless of water status, *B. cereus* inoculation induced a significant increase in IAA, gibberellins, and zeatin contents in *J. regia*. Under well-watered and light stress conditions, the abscisic acid content of walnut was significantly increased by *B. cereus* inoculation. Additionally, *B. cereus* inoculation significantly promoted the growth of plants in terms of ground diameter and plant height. As a result, PGPR inoculation could improve the drought resistance of *J. regia* and improve its photosynthetic characteristics and growth, suggesting that it is a useful supplementary measure for use in afforestation in arid and semiarid environments.

**Keywords:** *Juglans regia* seedlings; drought condition; PGPR; endogenous hormones; physiological and ecological characteristics



## 1. Introduction

Climate change is currently posing major challenges to agriculture at a time when the increasing world human population is placing more pressure on global food supplies [1]. Climate change will result in increased drought stress and water scarcity, which are already the abiotic stressors that are the most limiting to crop production [2]. In recent years, improving plant adaptability to drought by inoculating plants with exogenous genes and artificially synthesized bacterial preparation has become a "hot topic" for research worldwide. Hong et al. [3] showed that inoculation with arbuscular mycorrhizal (AM) fungi could significantly enhance the drought resistance of *Alpinia sinensis* seedlings. Zhang et al. [4] demonstrated that inoculation with AM fungi significantly improved the morphological and structural characteristics of the root system of *Sagittaria lamb* seedlings, which facilitated their adaptation to drought conditions. Some other studies examined the growth and water use efficiency of maize under drought stress by inoculating it with phosphorus-solubilizing

bacteria [5]. Although considerable efforts have been made toward the improvement of plant drought resistance by inoculation with microorganisms [3–6], relatively few studies have reported the screening of rhizosphere soil for plant growth-promoting rhizobacteria (PGPR) and their application in improving the drought resistance of plants, such as walnut.

Plant growth-promoting rhizobacteria are rhizosphere inhabitants that can promote plant growth and suppress diseases [7–10]. They are applied to many agronomically important crop plants for their beneficial effects, such as enhanced germination, increased nutrient availability, improved plant growth and crop yield, and reduced susceptibility to diseases [11–13]. It has been shown that the 1-aminocyclopropane-1-carboxylate (ACC) deaminase produced by PGPR can degrade the precursor of the endogenous plant hormone ethylene, ACC, to enhance the drought tolerance of wheat [14]. Timmusk and Wagner [15] were the first to show that inoculation with *Paenibacillus polymyxa* confers drought tolerance in *Arabidopsis thaliana* through the induction in the expression of the drought-responsive gene ERD15. Inoculation with *Bacillus amyloliquefaciens* 5113 and *Azospirillum brasilense* NO40 significantly alleviated the deleterious effects of drought stress on wheat [16]. Moreover, it has been suggested that PGPR could act as "biofertilizers" by helping plants tolerate abiotic stressors, such as drought or high salinity [17]. However, little information is available on the effects of PGPR on the growth and physiological and ecological characteristics of walnut under drought stress.

Walnut (*Juglans regia*) is an important nut and woody oil tree, and it has great potential for use in the greening of arid and barren mountainous areas, roads, and urban regions, together with the construction of farmland–forest networks [18]. If walnuts encounter drought stress in the process of growing, their growth and commerciality will be reduced, and walnuts could make adaptive adjustment responses to specific natural environmental conditions [19]. Li et al. [20] had suggested that "Ziren" walnut would respond to drought stress by enhancing the activities of superoxide dismutase (SOD) and peroxidase (POD) in the leaves. Additionally, the contents of soluble sugar, soluble protein, and free proline in the leaves of Xinjiang wild walnut seedlings showed an increasing trend under drought stress [21]. In this study, we investigated the effects of PGPR inoculation on the physioecological characteristics and growth of walnut seedlings under drought stress. We hypothesized that PGPR inoculation would be beneficial to walnut seedlings by improving their physiological and ecological characteristics and increasing their growth under drought stress conditions. The objectives of this study were to determine the feasibility of implementing PGPR inoculation in walnut cultivation in arid and semiarid regions and explore the mechanisms of drought resistance facilitated by PGPR.

## 2. Materials and Methods

### 2.1. Bacterial Screening and Identification

A bacterial strain was isolated and purified from the walnut rhizosphere that produced indoleacetic acid (IAA) and cytokinins at high levels, which was then preserved in the China General Microbiological Culture Collection Center (CGMCC No. 7069, Beijing, China). The trans-zeatin and IAA production by this bacterial isolate were 12.17 $\mu g/(mL \cdot OD600)^{-1}$ and 393.63 ng/mL, respectively, as determined by ultraperformance liquid chromatography–electrospray ionization tandem mass spectrometry [22], indicating the potential for the selected PGPR strain to increase plant growth. Based on the 16S rRNA sequencing data, the bacterial isolate showed 98.9% similarity with *Bacillus cereus* (AJ276351.1). The bacterial strain was indeed identified as *B. cereus* based on the comparison of its biochemical characteristics (Table 1) to those described in Bergey's Manual of Determinative Bacteriology [23] (Supplementary Information S1). The 16S RNA gene sequence of this isolate was submitted to GenBank under Accession Number KC428751 (Supplementary Information S2).

**Table 1.** Biochemical characteristics of the isolated *Bacillus cereus* L90.

| Gluten Hydrolysis | Starch Hydrolysis | Indole Test | V-P Test | Nitrate Reduction | Mannitol Test | Peroxidase Test | Lecithinase Test | Gas Production of Glucose | Acid Production of Glucose | Citrate Utilization |
|---|---|---|---|---|---|---|---|---|---|---|
| + | + | + | + | + | − | + | + | − | + | + |

+: positive; −: negative.

### 2.2. Basic Soil and Plant Materials

Our experiment was carried out in the greenhouse of Shandong Academy of Forestry, Shandong, China. The experimental soil was collected from local topsoil. Firstly, we removed stones, root stubble, sundries, etc., and then sieved the soil (3 mm aperture) after air drying, mixed the soil thoroughly, and took a part of the basic soil samples for the determination of basic soil properties. The available nitrogen (N), phosphorus (P), and potassium (K) content of the soil was 28.91, 27.08, and 80.50 mg·kg$^{-1}$, respectively, and the organic matter content was 7.36 g·kg$^{-1}$. The soil pH was 8.07 (as measured in a 1:2.5 soil: water suspension). Annual walnut seedlings developed by the Shandong Academy of Forestry and grown in nonwoven containers were used as plant materials in this study, and their average ($\pm$standard deviation) ground diameter and plant height were 2.43 $\pm$ 0.27 mm and 15.30 $\pm$ 0.39 cm, respectively. Plastic pots with a height of 20 cm and width of 30 cm were used in this study.

The pot experiment started on 16 April 2018, and followed a full factorial (4 $\times$ 2) design with 4 water statuses and 2 PGPR inoculations. Each pot was filled with 8.50 $\pm$ 0.20 kg of dry soil and six replicates were used for each experimental treatment.

### 2.3. Experimental Design

Drought stress treatments were applied on 18 June 2018. Each pot was fully watered prior to treatment to keep the initial soil moisture content consistent. After irrigation, the drought treatments were applied after the soil had naturally dried, and the field moisture capacity was 32.71% at that point. Four drought treatments were used in this experiment, including a control with normal watering conditions (CK, field moisture capacity = 60–70%) and treatments with light drought (LD, field moisture capacity = 50–60%), medium drought (MD, field moisture capacity = 40–50%), and severe drought stress conditions (SD, field moisture capacity = 30–40%).

The isolated PGPR strain was inoculated in beef extract peptone medium (0.3% (*w/w*) beef extract, 1% (*w/w*) peptone, 0.5% (*w/w*) sodium chloride, and 2% (*w/w*) agar; pH = 7.0–7.2) and cultured for 2 days at 37 °C under centrifugation at 180 r·min$^{-1}$. The zymotic fluid was centrifuged at 6000 r·min$^{-1}$ for 5 min, and the pellet was then rinsed with sterile saline 3 times before resuspending it in the inoculum. Twenty milliliters of the bacterial suspension was diluted to 500 mL with water, and the roots of walnut seedlings were then soaked in the diluted suspension before planting, with the remainder of the suspension then poured into the rhizosphere soil. Two inoculation treatments, one with PGPR inoculation and the other using irrigation with the same volume of saline (NS), were performed in combination with each of the four different water treatments. The soil moisture content in the pots was determined at 09:00 h every morning using an HH2 moisture meter (Theta probe type ML2X, Delta-T-devices, Cambridge, UK). To keep the soil moisture content within the range of each of the intended drought treatments, water was added until the moisture content reached the higher limit of the drought treatment if the moisture content was lower than the lower limit of that treatment. One hundred twenty days after the drought stress was applied, the aboveground and underground parts of the seedlings were harvested separately, and the levels of plant endogenous hormones and other physiological and ecological indicators were immediately measured after weighing each of these parts.

*2.4. Data Collection and Determinations*

Photosynthetic characteristics measurements were made using the mature leaves on the sunward side of each seedling. A portable gas exchange system (LI-6400, LI-COR, Lincoln, NE, USA) was used to determine the net photosynthetic rate ($P_n$) and stomatal conductance ($g_s$) from 09:00 to 11:00 h in the morning.

Endogenous hormone levels in the leaves: The methods used herein for the extraction and purification of the auxin IAA, the cytokinin zeatin (ZT), gibberellin (GA), and abscisic acid (ABA) were modified from those described by Bollmark et al. [24] and He [25].

Relative water content (RWC) was determined using the last expanded leaf according to the method described by Barrs and Weatherley [26], based on the following formula: RWC% = (fresh weight − dry weight) × 100/(turgor weight − dry weight).

Free proline was extracted from 1 g of fresh tissue in 5% ($w/v$) sulfosalicylic acid. The proline content was estimated by spectrophotometric analysis of the ninhydrin reaction at 515 nm, following the methods of Bates et al. [27].

The total content of soluble sugars in leaf extracts [28] was determined by the phenol sulfuric acid method [29].

Total superoxide dismutase (SOD) activity was measured on the basis of SOD's ability to inhibit the reduction of nitroblue tetrazolium (NBT) by photochemically generated superoxide radicals [30]. One unit of SOD activity was defined as the amount of enzyme required to inhibit the reduction rate of NBT by 50% at 25 °C. Catalase (CAT) activity was measured as described by Aebi [31]. The consumption of $H_2O_2$ (with an extinction coefficient of 39.6 $mM^{-1}cm^{-1}$) was monitored at 240 nm for 1 min. The reaction mixture consisted of 50 mM phosphate buffer (pH = 7.0) containing 10 mM $H_2O_2$ and 100 μL of cell extracts in a 2 mL volume.

The ground diameter and plant height of walnut seedlings were determined using a Vernier caliper and ruler with 0.5-mm accuracy at the end of the experiment (16 October 2018).

*2.5. Statistical Analyses*

The data were compared statistically among treatments using analysis of variance (ANOVA) in SPSS software (version 22.0). Each treatment was analyzed with at least six replicates, and all data are expressed herein as means ± the standard deviation (SD) of six replicates.

**3. Results**

*3.1. Physioecological Characteristics Measured in Leaves*

As shown in Table 2, regardless of whether walnut seedlings were inoculated with PGPR or not, the activities of antioxidant enzymes and the content of osmotic adjustment substances in them increased as the drought stress intensity increased, while the relative water content decreased gradually. Compared with the CK control plants, the activities of SOD and CAT, and the content of proline and total soluble sugars, increased by 148.92, 59.79, 122.07, and 112.72%, respectively, in PGPR-inoculated plants under severe drought conditions, while in plants treated with saline irrigation under severe drought conditions these indices were elevated by 99.13, 31.46, 160.27, and 169.08%, respectively. Additionally, under severe drought conditions, the relative water content in PGPR-treated and saline-treated plants was 17.43 and 20.25% lower than that in normal watering conditions, respectively. Therefore, PGPR inoculation did not alter the overall trends in antioxidant enzyme activities, osmotic regulator content, and relative water content in the leaves of walnut seedlings in response to changes in drought intensity.

**Table 2.** Effects of different drought and inoculation treatments on the antioxidant enzyme activities, osmotic adjustment substance content, and relative water content of walnut seedlings (Mean ± SD).

| Drought Treatment | Inoculation Treatment | Antioxidant Enzyme Activity ($U·g^{-1}$ FW) | | Osmotic Adjustment Substance Content ($mg·g^{-1}$ FW) | | Relative Water Content (%) |
|---|---|---|---|---|---|---|
| | | SOD | CAT | Proline | Soluble Sugars | |
| CK | PGPR | 347.63 ± 20.09 f | 73.16 ± 2.57 d | 16.49 ± 5.33 d | 6.37 ± 0.68 d | 87.11 ± 0.96 a |
| | NS | 353.02 ± 12.96 f | 69.87 ± 5.93 d | 18.25 ± 2.78 d | 6.21 ± 1.80 d | 87.64 ± 1.05 a |
| LD | PGPR | 449.95 ± 25.87 d | 82.90 ± 3.71 c | 18.68 ± 3.09 d | 6.55 ± 1.22 d | 84.03 ± 1.37 b |
| | NS | 392.26 ± 17.23 e | 73.39 ± 4.06 d | 20.13 ± 2.28 d | 6.86 ± 0.98 d | 83.56 ± 0.65 b |
| MD | PGPR | 687.54 ± 21.63 b | 95.76 ± 5.32 b | 26.38 ± 1.95 c | 9.32 ± 0.79 c | 78.45 ± 1.53 c |
| | NS | 571.60 ± 38.05 c | 82.03 ± 2.81 c | 27.59 ± 2.62 c | 12.96 ± 0.51 b | 75.31 ± 1.16 d |
| SD | PGPR | 865.31 ± 41.54 a | 116.90 ± 8.65 a | 36.62 ± 5.10 b | 13.55 ± 0.92 b | 71.93 ± 0.72 e |
| | NS | 702.98 ± 49.16 b | 91.85 ± 3.16 b | 47.50 ± 3.25 a | 16.71 ± 1.54 a | 69.89 ± 0.97 f |

Note: CK: normal watering; LD: light drought stress; MD: medium drought stress; SD: severe drought stress; SOD: superoxide dismutase; CAT: catalase. Different letters indicate significant differences among treatments at $p < 0.05$ by LSD.

With normal watering, no significant difference was observed in the activities of antioxidant enzymes between PGPR-inoculated and saline-irrigated plants. However, under light, moderate, and severe drought stress conditions, the SOD activity was increased by 14.71, 20.28, and 23.09% more, respectively, with PGPR inoculation than with saline irrigation, and the CAT activity was improved by 12.96, 16.74, and 27.27%, respectively. With normal watering and light drought stress, there were no significant differences in levels of osmotic adjustment substances and relative water content between the PGPR inoculation and saline irrigation treatments, but as the drought stress intensified, the difference between these became significant. The total soluble sugar content with PGPR inoculation was significantly lower than that with saline irrigation, whereas the relative water content was significantly higher with PGPR inoculation than that in the saline irrigation treatment under moderate and severe drought stress conditions. Thus, no significant effect of PGPR on the antioxidant enzyme activities, osmotic adjustment substance levels, and relative water content of walnut leaves was observed under normal watering conditions, but PGPR significantly improved the antioxidant enzyme activities and relative water content, and reduced the accumulation of osmotic adjustment substances, in walnut seedlings as the drought stress intensity increased.

### 3.2. Endogenous Hormone Levels in Leaves

As the drought stress intensity increased, the content of IAA, GA, and ZT in the leaves of walnut seedlings decreased gradually (Table 3). The content of IAA, GA, and ZT in the PGPR-treated seedlings under severe drought stress was reduced by 38.34, 28.68, and 23.73% compared to that under normal watering conditions, respectively, while the content of these hormones was decreased by 46.87, 38.24, and 32.75%, respectively, in the seedlings treated with saline irrigation. Regardless of water status, the content of IAA, GA, and ZT was higher with PGPR inoculation than that in plants treated with saline irrigation, and such differences became more significant as the drought stress intensity increased. Moreover, for the seedlings with saline irrigation, the ABA content progressively increased as the drought stress intensified, and the ABA content in seedlings under light, moderate, and severe drought stress conditions was 5.56, 53.70, and 75.93% higher than that under normal watering conditions, respectively. However, with RGPR inoculation, no significant difference in ABA content was observed between normal watering and light drought stress conditions, but the ABA content was significantly reduced under severe drought stress conditions. Additionally, the ABA content of the PGPR-inoculated group was dramatically higher than that of the saline-irrigated group with normal watering and light drought stress, which was the opposite of the patterns observed with severe drought stress.

**Table 3.** Effects of different drought and inoculation treatments on the content of different endogenous hormones in the leaves of walnut seedlings (Mean ± SD).

| Drought Treatment | Inoculation Treatment | IAA | GA | ZT | ABA |
|---|---|---|---|---|---|
| CK | PGPR | 30.65 ± 0.81 a | 180.34 ± 5.26 a | 3.16 ± 0.05 a | 0.72 ± 0.02 c |
|  | NS | 26.71 ± 1.03 b | 168.05 ± 3.53 b | 2.87 ± 0.05 b | 0.54 ± 0.03 e |
| LD | PGPR | 25.98 ± 1.45 b | 177.52 ± 3.91 a | 3.09 ± 0.07 a | 0.73 ± 0.02 c |
|  | NS | 23.87 ± 0.92 bc | 156.81 ± 6.05 c | 2.65 ± 0.09 c | 0.57 ± 0.05 e |
| MD | PGPR | 22.36 ± 0.79 c | 141.96 ± 5.70 d | 2.68 ± 0.10 c | 0.80 ± 0.03 b |
|  | NS | 19.22 ± 1.63 d | 126.57 ± 8.13 e | 2.39 ± 0.07 d | 0.83 ± 0.06 b |
| SD | PGPR | 18.90 ± 0.57 d | 128.62 ± 6.94 e | 2.41 ± 0.16 d | 0.66 ± 0.02 d |
|  | NS | 14.19 ± 1.28 e | 103.79 ± 8.68 f | 1.93 ± 0.22 e | 0.95 ± 0.03 a |

Note: CK: normal watering; LD: light drought stress; MD: medium drought stress; SD: severe drought stress; IAA: indoleacetic acid; ZT: zeatin; GA: gibberellin; ABA: abscisic acid. Different letters indicate significant differences among treatments at $p < 0.05$ by LSD.

### 3.3. Photosynthetic Characteristics

Changes in photosynthetic performance directly reflect the damage inflicted to plants by drought stress. As shown in Figure 1, both the $P_n$ and $g_s$ in the leaves of walnut seedlings decreased as the drought stress intensity increased, regardless of PGPR inoculation. The $P_n$ with PGPR inoculation and saline irrigation under severe drought stress conditions was 27.42 and 35.14% lower, respectively, than that under normal watering conditions, and the $g_s$ was reduced by 21.92 and 31.38% in them, respectively. Meanwhile, the difference in $P_n$ and $g_s$ between PGPR-inoculated and saline-irrigated plants was not significant under normal watering and light drought stress conditions, but the $P_n$ and $g_s$ of PGPR-inoculated seedlings became significantly higher than those in seedlings treated with saline irrigation under moderate and severe drought stress conditions, suggesting that the suppression by PGPR on the reduction in $P_n$ and $g_s$ by drought gradually increased as the drought stress intensity increased.

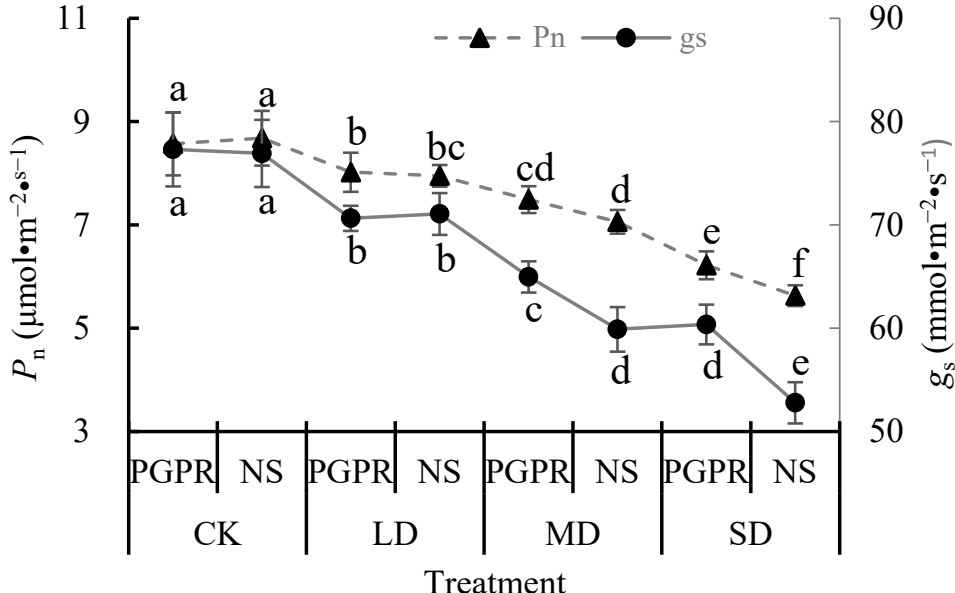

**Figure 1.** Effects of different drought and inoculation treatments on the $P_n$ and $g_s$ in the leaves of walnut seedlings. Bars are means, and error bars are standard deviations (n = 6). $P_n$: net photosynthetic rate; and $g_s$: stomatal conductance; CK: normal watering; LD: light drought stress; MD: medium drought stress; SD: severe drought stress. Different letters indicate significant differences among treatments at $p < 0.05$ by LSD.

*3.4. Ground Diameter and Plant Height*

The ground diameter and plant height of walnut seedlings tended to decrease as the drought stress intensity increased, regardless of PGPR inoculation (Figure 2). Under severe drought stress conditions, PGPR inoculation resulted in a 17.39 and 12.07% reduction in ground diameter and plant height, respectively, whereas saline irrigation led to decreases in these of 29.58 and 24.99%, respectively, compared with those in normal watering conditions. Regardless of water status, PGPR inoculation facilitated the growth of walnut seedlings to different degrees. Compared with saline irrigation, the ground diameter in PGPR-inoculated seedlings increased by 9.66, 19.92, 24.23, and 28.65% in the control, light, moderate, and severe drought stress conditions, respectively, and the plant height increased by 10.38, 15.14, 17.59, and 29.39%, respectively. These results indicated that PGPR could both significantly promote increases in the ground diameter and plant height of walnut seedlings and alleviate the damage caused by drought stress in them, and such effects were substantially enhanced when the drought stress was intensified.

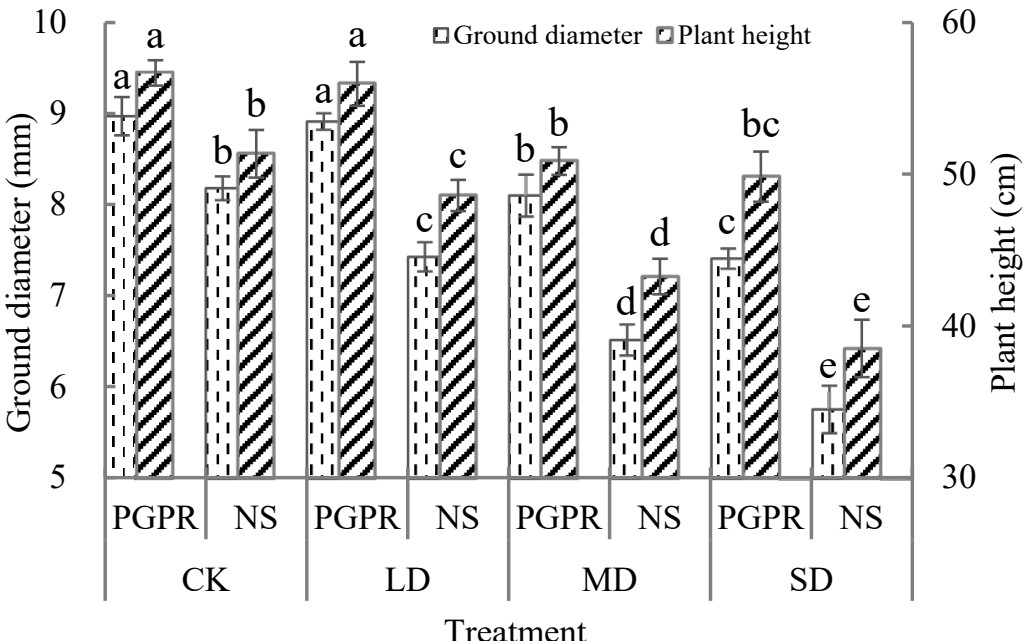

**Figure 2.** Effect of different drought and inoculation treatments on the ground diameter and plant height of walnut seedlings. Bars are means, and error bars are standard deviations (n = 6). CK: normal watering; LD: light drought stress; MD: medium drought stress; SD: severe drought stress. Different letters indicate significant differences among treatments at *p* < 0.05 by LSD.

**4. Discussion**

*4.1. Regulatory Enzymes in Leaves*

Some previous studies have demonstrated that plants produce a large number of reactive oxygen species when experiencing adverse conditions, but the damage caused by these can be prevented by increasing or initiating the activity of antioxidant enzyme systems, in which SOD and POD are two of the major enzymes [32–34]. The higher the activities of these enzymes are, the stronger their ability to scavenge free radicals will be, leading to plants developing stronger drought resistance [35]. In the present study, there was no significant difference in antioxidant enzyme activity between seedlings inoculated with *B. cereus* bacteria and those irrigated with saline under normal watering conditions, indicating that this strain of PGPR had no obvious effect on the enzymatic protection systems of walnut seedlings under these conditions. However, under the three drought stress conditions tested, the enzyme activity was significantly higher in plants with *B. cereus* inoculation than in those with saline irrigation, which suggested that *B. cereus*

accelerated the activation of the active oxygen scavenging systems in plants under drought stress conditions, thereby reducing the damage caused by the lipid peroxidation of cell membranes and improving the seedlings' drought resistance.

Osmotic adjustment is an important physiological mechanism used by plants to adapt to drought stress [36]. Under drought conditions, plants can maintain normal cell physiological processes by increasing intracellular solute concentrations, reducing osmotic potential, and maintaining a certain turgor pressure [37]. Proline and soluble sugars are important osmotic regulators in plants, and their accumulation under drought stress has been confirmed in many plant species [36,38]. Feng et al. [39] showed that the soluble sugar content of *Eucalyptus urophylla* and *Liquidambar formosana* reached its peak under light drought stress but decreased under moderate stress. In our study, the contents of proline and soluble sugars in walnut seedlings increased continuously as the drought stress intensity increased, with no such peak observed, which may be related to differences in the adaptive metabolic regulation mechanisms used in the leaves of different kinds of plants under drought stress. Ma and Liu [40] proposed the theory of "time difference", which postulates that a different limitation to stress tolerance exists in each different tree species. Compared with *Eucalyptus urophylla* and *Liquidambar formosana*, the tolerance limit of walnut seedlings found herein was substantially higher, which may have been one of the reasons for the absence of a peak in the accumulation of osmotic regulators in this study. Moreover, the proline produced under conditions of water stress has a role in pulling higher amounts of water into the plant [41]. Kohler et al. [42] reported that when plants were inoculated with PGPR, an increase in the quantity of proline in them was observed, whereas we found that PGPR inoculation reduced the proline content in our study. These differences can be attributed to walnut seedlings having stronger drought resistance than other species and may also be related to the use of different PGPR types, test periods, and other factors. Additionally, we also confirmed that the relative water content in the leaves of walnut seedlings decreased as the drought stress intensity increased, which is consistent with the results of previous studies [43].

### 4.2. Endogenous Hormones in Leaves

Endogenous hormones promote plant growth, and their content in a plant may change to coordinate physiological activities when plants are placed under adverse or stressful conditions [44]. In the present study, regardless of water status, PGPR inoculation significantly increased the content of ZT in leaves, which is mainly synthesized in the root meristem. PGPR can promote plant growth, especially by increasing the proportional abundance of hairy roots to enhance root absorption activity [8], which may be one of the reasons for the large increase in ZT content caused by PGPR inoculation herein. We showed that the growth-promoting effect of PGPR was strengthened when the drought stress intensified, and this may have been due to the increasing difference between photosynthate accumulation and consumption as the drought stress intensity increased. Furthermore, the content of IAA and GA also showed a similar pattern and increased with the drought intensity, suggesting that there was a balanced feedback relationship between ZT and IAA or GA. Regardless of water status, the content of IAA, GA, and ZT in the PGPR inoculation treatment were higher than those of plants in the saline irrigation treatment, and these differences were amplified gradually as the drought stress intensity increased.

ABA is a plant hormone with important regulatory functions in plant growth, stress resistance, stomatal movement, and gene expression [45]. In our study, the ABA content gradually increased in the saline irrigation treatment as the drought stress intensity increased, which is a common physiological regulation mechanism used by plants to adapt to drought conditions [46,47]. Under drought stress, the water status of cells is changed within a specific time frame, inducing the synthesis of ABA to effectively promote stomatal closure or partial closure, which reduces transpiration to improve water use efficiency [46]. The ABA content in PGPR-inoculated seedlings was significantly increased under normal watering and light drought stress conditions but was markedly reduced under severe

drought stress. Under normal environmental conditions, PGPR can promote the growth of the root system. ABA was mainly produced by the roots, which resulted in there being a higher ABA content in the PGPR inoculation treatment than that in the saline irrigation treatment under normal watering and light drought stress conditions. However, as the drought stress intensified, the growth of the root system was dramatically inhibited, leading to a significant reduction in the ABA content under severe drought stress conditions.

### 4.3. Photosynthetic Characteristics and Walnut Seedling Growth

Photosynthesis is the physiological basis of plant growth and can therefore reflect the growth potential and drought resistance of plants [48]. A large body of evidence has been accumulated showing that drought stress causes stomatal closure and reductions in $g_s$, leading to decreases in $P_n$ [48–50], which was further verified in the present study. We also illustrated that the alleviatory effect of PGPR on the $P_n$ and $g_s$ repression caused by drought was gradually strengthened as the drought stress intensity increased, indicating that the stronger the drought stress is, the greater the promoting effect of PGPR inoculation on $P_n$ and $g_s$ is. The main reason for this is that the bacteria with which seedlings were inoculated in this study could secrete cytokinins, which significantly increased the ZT content in the leaves of these walnut seedlings, and ZT can promote stomatal opening and thus increase $g_s$ and $P_n$. This facilitates the accumulation of photosynthates and stimulates the growth of the aboveground parts of the plant. In our study, inoculation with PGPR improved the growth of walnut seedlings in terms of their ground diameter and plant height under normal watering conditions, and this growth-promoting effect of PGPR was enhanced as the drought stress intensified. However, different environmental conditions, such as meteorological conditions, soil properties, or moisture, can also affect the growth and reproduction of microorganisms. Therefore, PGPR will only play their growth- and stress resistance-promoting roles when they can colonize the rhizosphere, meaning that the adaptability of PGPR to the soil environment is the key mediator of whether their functions can be realized [8]. Therefore, the strain of PGPR screened from the relatively drought-prone environment examined in this study is more likely to adapt to arid environments and thus to impact the photosynthetic capacity and aboveground growth of walnut seedlings under more arid conditions.

### 5. Conclusions

The results of this study show that *B. cereus* inoculation increased the antioxidant enzyme activities in leaves of walnut seedlings and changed their photosynthetic characteristics, whereas levels of osmotic adjustment substances were reduced because of PGPR inoculation with the increase in drought stress intensity. Regardless of water status, the contents of indoleacetic acid, gibberellins, and zeatin in leaves of *J. regia* were obviously increased as a result of *B. cereus* inoculation. Additionally, *B. cereus* inoculation dramatically promoted the growth of ground diameter and plant height of *J. regia*. These results suggest that PGPR inoculation could effectively alleviate the damage of drought stress to *J. regia* by regulating the antioxidant enzyme, osmotic adjustment substance, and photosynthetic characteristics, and stimulating some endogenous hormone production, showing a real potential to perform as a supplementary measure for the afforestation in arid and semiarid regions.

**Supplementary Materials:** The following supporting information can be downloaded at: https://www.mdpi.com/article/10.3390/agronomy13020290/s1, Supplementary Information S1: Preservation certificate of *B. cereus* used in this study; Supplementary Information S2: The 16S RNA gene sequence of *B. cereus* used in this study.

**Author Contributions:** F.L.: original draft preparation and bacterial screening and identification; H.M.: analysis of data and funding for publication; F.L. and B.L.: experiment conception and design, draft submission and revision; F.L. and Z.D.: data analysis; B.M. and D.J.: reagents, materials, and analysis tools contribution. All authors have read and agreed to the published version of the manuscript.

**Funding:** This work was funded by the National Natural Science Foundation of China (Grant No. 31570614).

**Data Availability Statement:** Not applicable.

**Acknowledgments:** We are grateful to the anonymous manuscript reviewers for their helpful comments and English corrections.

**Conflicts of Interest:** The authors declare no conflict of interest.

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
