# Peer review of "Effects of Plant Growth-Promoting Rhizobacteria on the Physioecological Characteristics and Growth of Walnut Seedlings under Drought Stress"

_agronomy, doi:10.3390/agronomy13020290_

Round 1
Reviewer 1 Report
Authors looked at how the physioecological traits and development of walnut seedlings under drought stress were influenced by PGPR. According to the authors, PGPR inoculation might increase a plant's capacity to survive drought conditions while also enhancing its photosynthetic traits and development, indicating that it is a valuable supplemental strategy for use in afforestation in arid and semi-arid environments.
I have some issues with the manuscript:
- Include some numeric data in the abstract.
- Keywords: Don't use the same words mentioned in the title again, and double-check the police size here.
- Drought stress and its impacts on plants can be discussed in more detail in the introduction section.
- Line 48 to 53: please add some relevant references here.
- Line 54: Avoid starting paraphrase with acronyms.
- Line 69: If you have any information about walnuts being sensitive to drought stress, please put it here.
- Line 98: Where you collected the experimental soil, your sampling procedure, and the depth are all missing information.
- Why did you begin the experiment on April 16 and apply drought stress two months later, on June 18?
- You said the trial was conducted using the RCBD design, yet the results show two factors (drought and inoculation treatments). Please verify that your RCBD, and not a split plot or another design, is the best one for this situation.
- Table 2: table must be self-explanatory. This means that all symbols and abbreviations need to be defined either in the table itself or in the notes under the table. Please check all tables.
- I believe the introduction may also be enhanced.
- Revise the list of references: there are some errors here (Lines: 396, 401, 406).
Author Response
Dear reviewer,
Thank you for your quick response concerning our manuscript entitled “Effects of plant growth-promoting rhizobacteria on the physioecological characteristics and growth of walnut seedlings under drought stress”. he comments helped us revise our manuscript, which we hope would meet with your and your reviewers’ approval.
The line numbers in the beginning of each sentence refer to the original manuscript. We listed the modifications as follows:
- Keywords: Don't use the same words mentioned in the title again, and double-check the police size here.
Reply: The original keywords were replaced by “Juglans regia seedlings; drought condition; PGPR; endogenous hormones; physiological and ecological characteristics”.
- Drought stress and its impacts on plants can be discussed in more detail in the introduction section.
Reply: Some discussion of walnuts and drought stress has been added in the introduction.
- Line 48 to 53: please add some relevant references here.
Reply: Relevant references [5] and [3-6] were added.
- Line 54: Avoid starting paraphrase with acronyms.
Reply: “PGPR” was replaced by“Plant growth promoting rhizobacteria”.
- Line 69: If you have any information about walnuts being sensitive to drought stress, please put it here.
Reply: Some discussion of walnuts and drought stress has been added in the introduction.
- Line 98: Where you collected the experimental soil, your sampling procedure, and the depth are all missing information.
Reply: The following sentence has been added: “The experimental soil was collected from local topsoil. Firstly, removed stones, root stubble, sundries, etc., then sieved the soil (3 mm aperture) after air drying, mixed the soil thoroughly, and took a part of the basic soil samples for the determination of the basic soil properties.”
- Why did you begin the experiment on April 16 and apply drought stress two months later, on June 18?
Reply: When walnut seedlings were transplanted into pots, the seedlings needed a process of adaptation and colonization. From years of experience in pot experiments, it could be known that about two months of colonization time was more appropriate.
- You said the trial was conducted using the RCBD design, yet the results show two factors (drought and inoculation treatments). Please verify that your RCBD, and not a split plot or another design, is the best one for this situation.
Reply: We are sorry that we made a mistake here.
The pot experiment was followed a full factorial (4×2) design with 4 water status and 2 PGPR inoculations.
- Table 2: table must be self-explanatory. This means that all symbols and abbreviations need to be defined either in the table itself or in the notes under the table. Please check all tables.
Reply: The definition of all symbols and abbreviations were added in the notes under the table.
- I believe the introduction may also be enhanced.
Reply: Some supplements and improvements have been made to the introduction, and 5 references were added.
- Revise the list of references: there are some errors here (Lines: 396, 401, 406).
Reply: The errors from the list of references were revised.
Special thanks to you for your quick and kindly comments.
Once again, thank you very much for your comments and suggestions.
Sincerely yours.
Fangchun Liu

Reviewer 2 Report
Article title:
“Effects of plant growth-promoting rhizobacteria on the physioecological characteristics and growth of walnut seedlings under drought stress".
The purpose of this study was to investigate the mechanisms of drought resistance made possible by PGPR and to assess the viability of PGPR inoculation in walnut agriculture in arid and semi-arid locations.
The work done is certainly of international interest and the format applied is certainly suitable for an article. This article dealt with the topic in a different and attractive way, and the titles are related to each other. The work is original, of particular interest, and can certainly stimulate research on this topic. The conclusion summarizes the aims of the work and future prospects
Regarding the manuscript, there are some points, that the authors should be to modify i.e.
Minor comments:
- Line 13: by water stress is an interesting alternative approach to mitigate the drought stress experienced by plants. (Rewrite again).
- Line 14-17: It is known that plant growth-promoting rhizobacteria (PGPR) can colonize plant roots and increase plant growth. However, less information is available on the effects of PGPR on the growth and physioecological characteristics of plants under drought stress, including walnut trees. (Delete).
- Lines 21, 24, 28, 29, 89 and 93: Bacillus cereus (Italic).
- Line 22 and 270: Little effect (Change little word).
- Line 27: Because of (Change it).
- Line 43: exogenous genes and synthetic bacteria. What's means?
- Line 61: Arabidopsis thaliana (Italic).
- Line 89: isolate (change to strain).
- Write the abbreviations for the treatments used in full below the tables and figures.
Author Response
Dear reviewer,
Thank you for your quick response concerning our manuscript entitled “Effects of plant growth-promoting rhizobacteria on the physioecological characteristics and growth of walnut seedlings under drought stress”. he comments helped us revise our manuscript, which we hope would meet with your and your reviewers’ approval.
The line numbers in the beginning of each sentence refer to the original manuscript. We listed the modifications as follows:
-Line 13: by water stress is an interesting alternative approach to mitigate the drought stress experienced by plants. (Rewrite again).
Reply: The sentence was replaced by “of water stress is an interesting alternative to alleviate the drought stress experienced by plants.”
-Line 14-17: It is known that plant growth-promoting rhizobacteria (PGPR) can colonize plant roots and increase plant growth. However, less information is available on the effects of PGPR on the growth and physioecological characteristics of plants under drought stress, including walnut trees. (Delete).
Reply: The sentences were deleted.
- Lines 21, 24, 28, 29, 89 and 93: Bacillus cereus (Italic).
Reply: The Bacillus cereus from the lines of 21, 24, 28, 29, 89 and 93 were indicated in italics.
- Line 22 and 270: Little effect (Change little word).
Reply: The“little” from the lines of 22 and 270 had been replaced by “no obvious”.
- Line 27: Because of (Change it).
Reply: The sentence was replaced by “Regardless of water status, B. cereus inoculation induced a significant increase of IAA, gibberellins, and zeatin contents in J. regia”.
- Line 43: exogenous genes and synthetic bacteria. What's means?
Reply: “exogenous genes and artificially synthesized bacterial preparation” was added.
-Line 61: Arabidopsis thaliana (Italic).
Reply: The “Arabidopsis thaliana” was indicated in italics.
-Line 89: isolate (change to strain).
Reply: The “isolate” was changed to “strain”.
- Write the abbreviations for the treatments used in full below the tables and figures.
Reply: The definition of all abbreviations were added in the notes under the tables and figures.
Special thanks to you for your quick and kindly comments.
Once again, thank you very much for your comments and suggestions.
Sincerely yours.
Fangchun Liu

Round 2
Reviewer 1 Report
Dear authors
I would like to thank you for taking the necessary time and effort to review the manuscript. I sincerely appreciate the revised version.